# Production of Indole Auxins by *Enterobacter* sp. Strain P-36 under Submerged Conditions

**Francesca Luziatelli** [1], **Francesca Melini** [1,2], **Paolo Bonini** [3], **Valentina Melini** [2], **Veronica Cirino** [4] **and Maurizio Ruzzi** [1,*]

1   Department for Innovation in Biological, Agro-Food and Forest Systems (DIBAF), University of Tuscia, 01100 Viterbo, Italy; f.luziatelli@unitus.it (F.L.); francesca.melini@crea.gov.it (F.M.)
2   Council for Agricultural Research and Economics (CREA), Research Centre for Food and Nutrition, I-00178 Rome, Italy; valentina.melini@crea.gov.it
3   Next-Generation Agronomics (NGA) Laboratory, 43762 Tarragona, Spain; paolo.bonini@atens.es
4   Atens—Agrotecnologias Naturales SL, 43762 Tarragona, Spain; veronica.cirino@atens.es
*   Correspondence: ruzzi@unitus.it; Tel.: +39-349-425-6659

**Abstract:** Bioactive compounds produced by plant growth-promoting bacteria through a fermentation process can be valuable for developing innovative second-generation plant biostimulants. The purpose of this study is to investigate the biotechnological potential of *Enterobacter* on the production of auxin—a hormone with multiple roles in plant growth and development. The experiments were carried in Erlenmeyer flasks and a 2-L fermenter under batch operating mode. The auxin production by *Enterobacter* sp. strain P-36 can be doubled by replacing casein with vegetable peptone in the culture medium. Cultivation of strain P36 in the benchtop fermenter indicates that by increasing the inoculum size 2-fold, it is possible to reduce the fermentation time from 72 (shake flask cultivation) to 24 h (bioreactor cultivation) and increase the auxin volumetric productivity from 6.4 to 17.2 mg [IAA$_{equ}$]/L/h. Finally, an efficient storage procedure to preserve the bacterial auxin was developed. It is noteworthy that by sterilizing the clarified fermentation broth by filtration and storing the filtrated samples at +4 °C, the level of auxin remains unchanged for at least three months.

**Keywords:** plant biostimulants; plant growth-promoting rhizobacteria (PGPR); *Enterobacter*; auxin (IAA); organic agriculture

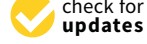



## 1. Introduction

Food production and consumption are crucial to any society, and great reliance is placed on farming activities. Current agriculture has, however, come to a crossroads and is now called to face the dual challenge of ending hunger and achieving food security, while promoting resilient agricultural practices and sustainable food production systems [1]. New technologies and approaches have been increasingly researched to pursue global food security and agricultural sustainability, and within this framework, the application of bio-based products such as plant biostimulants emerged as a promising alternative to agrochemicals and a new frontier of investigation [2].

Plant biostimulants are defined by European legislation as products "stimulating plant nutrition processes with the sole aim of improving (a) nutrient use efficiency; (b) tolerance to abiotic stress; (c) quality traits, and; (d) availability of confined nutrients in soil or rhizosphere" [3]. Plant biostimulants can be microbial or non-microbial [3], and encompass humic substances (e.g., humic and fulvic acids), vegetal- or animal-based protein hydrolysates, seaweed extracts, and botanicals, arbuscular mycorrhizal fungi, and plant growth-promoting rhizobacteria (PGPR) [4,5].

PGPR are endophytes and epiphytic bacteria, inhabiting the rhizosphere and the rhizoplane, that act as biostimulants in agriculture [5]. They can increase crop tolerance against abiotic stresses and improve nutrient use efficiency, plant health, productivity, and

yield at different stages [6–8]. They exert the aforesaid beneficial effects by direct and indirect mechanisms [5].

Biosynthesis of plant growth regulators, including auxins, gibberellins, cytokinins, and abscisic acid, is among the direct mechanisms correlated to the stimulation of plant growth and nutrient acquisition. Microbial regulators modulate plant hormone levels in plant tissue and have been found to have effects that are similar to exogenous phytohormone application [9].

Indole-3-acetic acid (IAA) is the most common plant hormone of the auxin class which regulates various processes of plant growth. This compound is produced by both plants and microorganisms, together with other auxin-related compounds [10]. The identification of the major bacterial producers of IAA and other auxin-related compounds has thus become pivotal in the approach to identify new paths for sustainable agriculture.

Strains belonging to the genera *Pseudomonas*, *Bacillus*, *Pedobacter*, *Pantoea*, *Luteibacter*, *Acinetobacter*, *Lysobacter*, and *Enterobacter* have been increasingly investigated for their plant growth-promoting traits and ability to produce indole auxins/IAA [11]. Production of bacterial auxins has proved, so far, to vary from strain to strain, and is affected by different factors, among which strain growth conditions and precursor availability, such as tryptophan [12]. Moreover, in the exometabolome of auxin-producing bacteria, strain-specific postbiotic metabolites that act in synergy with auxin and accelerate the adventitious rooting formation can be present [13,14].

Within this framework, it is crucial to design suitable growth conditions for the large-scale production of indole auxins/IAA. In particular, precise and careful monitoring and control of culture conditions by the use of fermenters could provide a better understanding of environmental factors controlling IAA biosynthesis.

The aim of this study was to investigate the biotechnological potential of *Enterobacter* on indole auxin production at shake flask and bench-scale bioreactor level evaluating the possibility to use a medium free of animal-derived ingredients. To that aim, we selected *Enterobacter* sp. strain P-36, a strong IAA producer [15], and investigated the effects of the growth medium, inoculum size, and incubation time on indole auxins/IAA production. In parallel, we evaluated the effect of pH, temperature, and sterilization procedures on the stability of crude preparations of indole auxin, to have a more defined view of the production process.

## 2. Materials and Methods

### 2.1. Bacterial Strain and Culture Media

*Enterobacter* sp. strain P-36 was originally isolated by a commercial mycorrhizal inoculum and described as a strong producer of IAA [15].

The strain was stocked in LB broth (Lennox) [16] containing glycerol 20% (*v/v*) at −80 °C and revived in LB broth at 30 °C under agitation [180 revolutions/min (rpm)].

For indole auxin (indole-3-acetic acid) production, three different media were investigated: (i) LB broth (Lennox) (per liter, tryptone 10 g, yeast extract 5 g, NaCl 5 g); (ii) a vegetal peptone—yeast extract (VY) medium, an alternative version of LB broth in which tryptone was replaced with 10 g/L of vegetal peptone from pea (a GMO-free alternative to veggietone soya peptone); (iii) a yeast extract sucrose medium (YES) containing $1 \times$ M9 salt solution, yeast extract (5 g/L) and sucrose (5 g/L) [13].

All media were amended with a filter-sterile solution of L-tryptophan (4 mM) as auxin precursor/inducer.

Tryptophan (Trp) was prepared as a 40 mM stock solution in mildly alkaline water (including NaOH), filter sterilized (0.22 μm), and stored in the dark at −20 °C.

Tryptone, yeast extract, and vegetal peptone were purchased from Thermo Fisher Scientific (Waltham, MA, USA); sucrose, tryptophan, and chemicals were purchased from Sigma-Aldrich (Merck KGaA, Darmstadt, Germany).

To decouple interferences related to culture vessels [17], all experiments were carried out in Erlenmeyer flasks compliant with DIN ISO 1773:1999.

### 2.2. Indole Auxin Production in Shake Flasks

Pre-seed cultures from a freeze glycerol stock of *Enterobacter* sp. strain P-36, were inoculated into 50 mL of LB broth in 500 Erlenmeyer flasks and incubated overnight at 180 rpm and 30 °C. Seed cultures in the late exponential phase of growth [optical density at 600 nm ($OD_{600}$) of 4.5–4.8] were used to inoculate 25 mL of tryptophan (4 mM) amended medium. The cultures (initial $OD_{600}$ of 0.2) were grown in 250 mL Erlenmeyer flasks, in triplicate, in an INFORS HT Multitron (Bottmingen, Switzerland) incubator at 180 rpm and 30 °C. After 24 h of growth, 10 mL of each culture was recovered for indole auxins quantification. To maintain a constant medium-to-flask volume ratio, 9 flasks were inoculated for each treatment and triplicates of these flasks were sacrificed at each time point.

### 2.3. Production of Indole Auxin in a 2-L Stirred Tank Fermenter

For the optimization of the indole auxin production, batch fermentation was carried out in a 2-L stirred tank fermenter connected to an Applikon ADI 1020 Bio Controller (Applikon Biotechnology, Delft, NL, USA). The bioreactor was equipped with two Rushton impellers, digital ISM probes for pH (Mettler-Toledo S.p.A., Milan, Italy) and dissolved oxygen (DO), and a platinum RTD (Pt100).

The bioreactor with the culture medium (working volume of 1.1 L) was autoclaved for 60 min at 121 °C. After autoclaving, a sterile tryptophan solution was added under controlled aseptic conditions to obtain a final concentration of 4 mM.

The growth was carried out in VY medium at 30 °C under aerobic conditions. Oxygen level was kept constant, higher than 20% air saturation, by increasing the agitation speed from 250 rpm (initial condition) to 500 rpm and by maintaining a constant airflow rate of 1.5 (vol/vol/min).

The initial pH of the medium was adjusted to 6.5 by the addition of 1 M HCl, and the growth was carried out without pH control.

The fermenter was inoculated with an appropriate volume of an LB culture ($OD_{600} \sim 6.3$) grown overnight in shake flasks at 180 rpm and 30 °C to have an initial cell density of 0.5 or $1 \times 10^9$ cells/mL (corresponding to a predicted initial $OD_{600}$ of 0.2 or 0.4). The fermentations were performed in duplicate.

### 2.4. Measurement of Indole-3-Acetic Acid (IAA) and Related Indolic Metabolites Production

Indole auxins were measured in a spent medium by colorimetric assay with Salkowski's reagent, according to the method by Patten and Glick [18]. Briefly, cell culture was centrifuged at 8000 rpm for 10 min in a multispeed centrifuge (Thermo Fisher Scientific, Waltham, MA, USA) and the cell pellet was discharged. An aliquot (1 mL) of filter-sterilized (0.22 μm) supernatant, diluted as needed, was added to 2 mL of Salkowski's reagent (0.5 M $FeCl_3$, 35% $v/v$ $HClO_4$). The mixture was incubated at room temperature for 20 min and absorbance of the developed pink color was read at 530 nm. Indole auxin concentration in the culture was determined by using a calibration curve of pure indole-3-acetic-acid (Sigma–Aldrich, Milan, Italy), as a standard. All these experiments were carried out in triplicate and values were expressed in mg of $IAA_{equ}$ per liter.

### 2.5. Evaluation of Auxin Stability

The effect of storage time and conditions on the stability of *Enterobacter* crude auxin preparations was also investigated. Cell cultures grown in the 2-L stirred tank fermenter for 48 h were recovered and centrifuged at 8000 rpm for 10 min. The pellet was discharged, as specified for auxin determination; supernatant was sterilized by filtration (Ø = 0.22 μm) or thermal treatment (121 °C for 20 min), optionally buffered at pH 9.0, and stored at +4 or −20 °C.

One aliquot (200 mL) was not treated (*U*, unbuffered, at pH 8.0 ± 0.1). A second aliquot was added with a NaOH (10 N) solution to adjust the pH to 9.0 (*B*, buffered, with pH 9). Two subsets were formed for each aliquot: one subset was sterilized by

filtration; another was treated thermally. Each subset ($U_{tt}$, unbuffered thermally treated; $U_f$, unbuffered sterilized by filtration; $B_{tt}$, buffered thermally treated; $B_f$, buffered sterilized by filtration) was stored at +4 °C and −20 °C. In this way, 8 different assay conditions were obtained, and each sample was tested at 30, 60, and 90 days to determine the residual auxin concentration. To avoid any effect by thawing, a suitable number of broth aliquots were prepared, so that each sample was thawed only once.

### 2.6. Metabolome Analysis by Quadrupole Time-of-Flight Liquid Chromatography-Mass Spectroscopy

For untargeted metabolomics, the sample (1 mL) was extracted in a cold acidified (0.1% HCOOH) 80/20 methanol/water mixture (5 mL) using an Ultra-Turrax Homogenizer (Ika T-25, Staufen, Germany), centrifuged at 1200 rpm, and filtered through a 0.2 μm cellulose membrane. The analysis was carried out on an Agilent 6550 Q-TOF (Madrid, Spain) with ESI source, coupled with an Agilent 1290 UHPLC equipped with a C18 column (100 × 2.1 mm internal diameter, 1.7 μm). The injection volume was 2 μL for all samples. A pooled quality control was obtained by mixing 10 μL of each sample and acquired in tandem mass spectroscopy mode using iterative function five consecutive times to increase the number of compounds with associated MS2 spectra. Blank filtering, alignment, and identification were accomplished as described elsewhere using MS-DIAL and MS-FINDER [13]. In addition, IAA = 3-Indoleacetic acid and IAM = Indole-3-acetamide standards (Sigma Aldrich, Madrid, Spain) were injected in the same chromatographic run to reach level 1 in compound identification as defined in the Metabolomics Standard Initiative [19]. All the other compounds must be considered as level 2. The table with all compound peaks heights was exported from MS-DIAL into MS-FLO, to reduce false positives and duplicates, and utilized for PLS-DA and chemical enrichment analysis as described by Luziatelli et al. [13].

### 2.7. Statistical Analysis

Statistical analysis was performed by the one-way analysis of variance using the SigmaStat 3.1 package (Systat Software Inc., San Jose, CA, USA).

## 3. Results

### 3.1. Production of Indole-3-Acetic Acid (IAA) and Related Indolic Metabolites in Shake Flasks

The first batch fermentation in a shake flask was carried to evaluate the biotechnological potential of *Enterobacter* sp. strain P-36 as a producer of indole auxins. In particular, the effect of growth time and medium on indole auxins production, biomass yield, and auxin-specific productivity at 30 °C and 180 rpm were investigated. Preliminary experiments on LB medium indicated that no significant auxin production was obtained without amending the medium with tryptophan. Therefore, all experiments were carried out with tryptophan (4 mM) as a precursor and inducer of the IAA biosynthetic pathway.

Results showed that the highest auxin/IAA level was obtained when strain P-36 was grown for 72 h, whichever medium was used (Figure 1A).

Comparing data of auxin/IAA production in the different growth media amended with tryptophan, it emerged that the highest levels occurred in VY and LB medium: $459.3 \pm 8.0$ and $212.4 \pm 2.6$ mg IAA$_{equ}$ L$^{-1}$, respectively. Interestingly, the use of a vegetable peptone (VY medium) determined a two-fold increase in the auxin level compared to animal-based peptone (LB medium; Figure 1). After 24 h of growth, the auxin/IAA level in VY medium ($219.4 \pm 10.4$ mg IAA$_{equ}$ L$^{-1}$) was comparable to the highest level obtained in LB medium ($212.4 \pm 2.6$ mg IAA$_{equ}$ L$^{-1}$) and about 1.8-fold higher than the one obtained in YES medium at 72 h ($123.9 \pm 8.1$ mg IAA$_{equ}$ L$^{-1}$; Figure 1A).

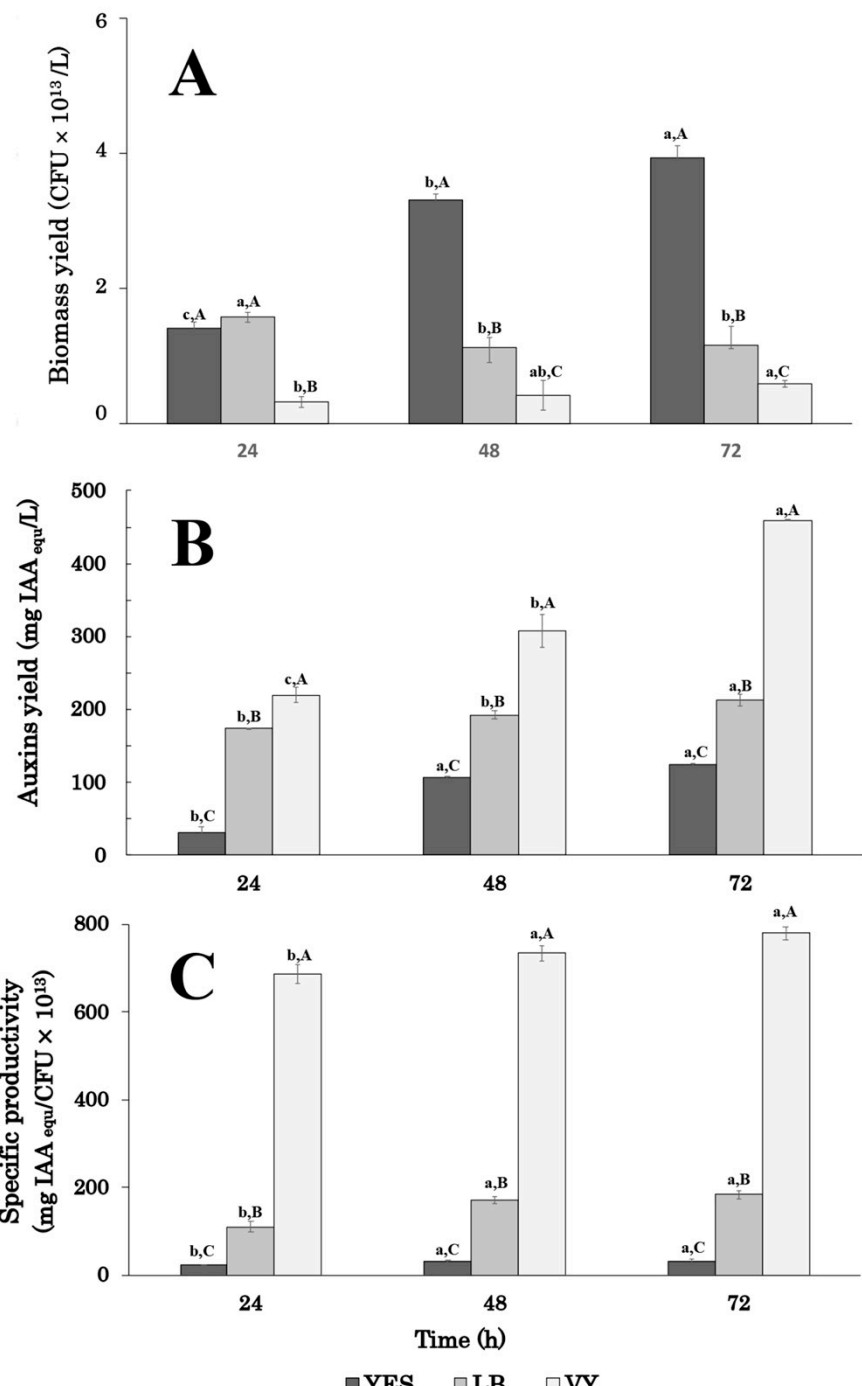

**Figure 1.** Effect of growth medium and incubation time on the indole auxin production yield (**A**), biomass yield (**B**), and auxin specific productivity (**C**) by *Enterobacter* sp. strain P-36 grown in shake flasks. Results are means $\pm$ SD of three independent experiments and auxin levels are expressed as IAA$_{equ}$. Differences in letters indicate that the values are significantly different ($p > 0.05$). Lower-case (a,b,c) letters are referred to values of the same series (growth medium), while capital (A,B,C) letters indicate statistically significant differences among values of different series (same incubation time).

Biomass and auxin production followed different patterns (Figure 1A,B). Between 24 and 72 h, the biomass yield increased by 2.8-fold (from $1.41 \pm 0.1 \times 10^{13}$ to $3.93 \pm 0.2 \times 10^{13}$ CFU L$^{-1}$) in YES medium and 1.8-fold (from $3.2 \pm 0.8 \times 10^{12}$ to $5.9 \pm 0.5 \times 10^{12}$ CFU L$^{-1}$) in VY medium, respectively. In contrast, on LB medium, which stimulates biofilm formation in thin layers, the biomass yield slightly decreased between 24 and 48 h, remaining constant up to

72 h (Figure 1B). The highest biomass yield ($3.9 \pm 0.2 \times 10^{13}$ CFU/L) was obtained in the YES medium after 72 h of growth. This value was about 3.4- and 6.6-fold higher than the one obtained in LB and VY, respectively (Figure 1B).

Data on auxin-specific productivity confirmed that the composition of the culture medium significantly affected the auxin production efficiency of P36 cells (Figure 1C). In all conditions, the highest auxin-specific productivity was obtained after 48 h of growth and remained constant through the 48- and 72-h period. On VY medium, the specific productivity was about 23- and 4-fold higher than in YES and LB medium, respectively (Figure 1C).

### 3.2. Auxin/IAA Production in 2-L Stirred Tank Fermenter

To better understand the link between the auxin/IAA biosynthesis and the main culture parameters, *Enterobacter* sp. strain P-36 was grown under controlled bioreactor conditions.

Based on the data obtained upon cultivation in shake flasks, the VY medium was identified as the best growth medium. This medium was thus used for the scale-up experiments of auxin/IAA production by *Enterobacter* sp. strain P-36.

To monitor the effect of the fermentation time and the size of the inoculum on auxin/IAA production, experiments were carried out in 2-L stirred tank bioreactor with a 1.1 L working volume of VY medium amended with tryptophan, 250–500 rpm agitation speed, and 20% oxygen level. The initial inoculum was set at 0.5 or $1.0 \times 10^9$ cells/mL (predicted $OD_{600}$ of 0.2 or 0.4), and indole auxins production was monitored from 5 up to 72 h of growth (Figure 2A,B).

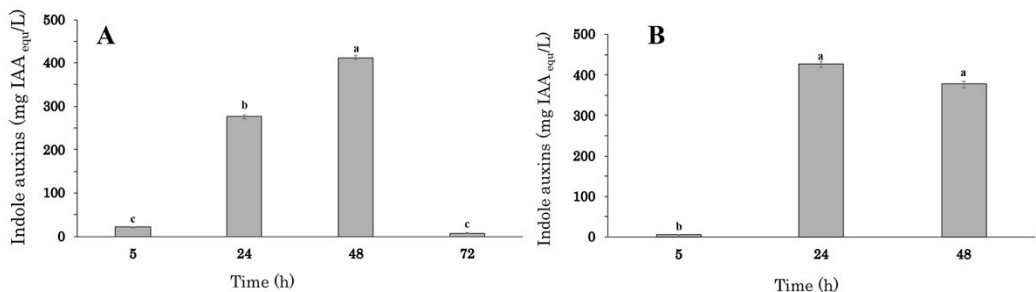

**Figure 2.** Auxin accumulation profile during batch cultures in VY + Trp medium inoculated at 0.5 (**A**) and $1.0 \times 10^9$ cell/mL (**B**). at of *Enterobacter* sp. P-36 inoculated. The error bars represent standard errors of the means (n = 3). Differences in letters indicate that the values are significantly different ($p > 0.05$).

At the lower inoculum concentration ($0.5 \times 10^9$ cell/mL), the highest auxin/IAA level ($412.5 \pm 4.9$ mg $IAA_{equ}$ $L^{-1}$) was recorded at 48 h and was 1.5-fold higher than the value observed at 24 h ($275.9 \pm 5.0$ mg $IAA_{equ}$ $L^{-1}$; Figure 2A). The application of fermentation times longer than 48 h determined a significant decrease in the auxin/IAA concentration ($7.7 \pm 1.5$ mg $IAA_{equ}$ $L^{-1}$).

Increasing the initial inoculum from 0.5 to $1.0 \times 10^9$ cell/mL, the highest auxin/IAA level ($427.9 \pm 8.1$ mg $IAA_{equ}$ $L^{-1}$) was achieved after 24 h of growth, and no significant decrease was observed in the subsequent 24 h (Figure 2B).

### 3.3. Development of a Protocol for Indole Auxin Storage

To evaluate the effect of pH, sterilization treatment, and storage temperature on the stability of auxin/IAA produced by *Enterobacter* sp. strain P-36 and develop a reliable protocol for the preservation of these metabolites, cells were separated from the exhausted growth medium, and aliquots of the latter were differentially treated as described in Methods. Eight different samples were prepared: buffered and unbuffered thermally treated samples stored at +4 °C [$B_{tt}$ (4) and $U_{tt}$ (4)] or −20 °C [$B_{tt}$ (20) and $U_{tt}$ (20)]; buffered and unbuffered filter-sterilized samples stored at +4 °C [$B_f$ (4) and $U_f$ (4)] or −20 °C [$B_f$ (20) and $U_f$ (20)].

At $T_0$, the auxin concentration was the same ($379.7 \pm 14$ mg $IAA_{equ}$ $L^{-1}$; Table 1) in both buffered and unbuffered samples, either sterilized by filtration or thermal treatment and stored at +4 °C or −20 °C.

**Table 1.** Effect of pH, sterilization method, and storage temperature on the stability of crude auxin/IAA preparations.

| | Auxin/IAA (mg $IAA_{equ}$ $L^{-1}$) | | | |
|---|---|---|---|---|
| **Sample** | $T_0$ | $T_{30}$ | $T_{60}$ | $T_{90}$ |
| Thermally treated | | | | |
| $B_{tt}$ (4) | $379.7 \pm 14$ [d,D] | $496.5 \pm 12$ [b,A] | $476.4 \pm 6.3$ [b,B] | $438.4 \pm 0.7$ [c,D] |
| $B_{tt}$ (−20) | $379.7 \pm 14$ [d,D] | $471.0 \pm 3.9$ [b,B] | $413.0 \pm 16$ [d,C] | $433.9 \pm 4.1$ [c,D] |
| $U_{tt}$ (4) | $379.7 \pm 14$ [d,D] | $536.4 \pm 1.3$ [a,A] | $457.3 \pm 13.7$ [b,B] | $452.2 \pm 4.9$ [b,B] |
| $U_{tt}$ (−20) | $379.7 \pm 14$ [d,D] | $536.5 \pm 1.5$ [a,A] | $463.8 \pm 9.3$ [b,B] | $457.6 \pm 0.7$ [b,B] |
| Filtered | | | | |
| $B_f$ (4) | $379.0 \pm 0.1$ [d,D] | $466.2 \pm 25.6$ [c,B] | $476.9 \pm 3.1$ [c,B] | $478.7 \pm 12.9$ [c,B] |
| $B_f$ (−20) | $379.0 \pm 0.1$ [d,D] | $494.2 \pm 1.8$ [b,A] | $449.5 \pm 9.4$ [c,B] | $437.6 \pm 6.9$ [c,B] |
| $U_f$ (4) | $379.0 \pm 0.1$ [d,D] | $549.2 \pm 1.7$ [a,A] | $460.1 \pm 1.9$ [c,B] | $465.0 \pm 7.8$ [c,B] |
| $U_f$ (−20) | $379.0 \pm 0.1$ [d,D] | $509.5 \pm 1.4$ [b,A] | $465 \pm 1.0$ [c,B] | $456.6 \pm 5.2$ [c,B] |

$T_0$: day 0; $T_{30}$: 30th day; $T_{60}$: 60th day; $T_{90}$: 90th day. $U_{tt}$ (4), unbuffered thermally treated sample stored at +4 °C; $U_{tt}$ (20), unbuffered thermally treated sample stored at −20 °C; $U_f$ (4), unbuffered filter-sterilized sample stored at +4 °C; $U_f$ (20), unbuffered filter-sterilized sample stored at −20 °C; $B_{tt}$ (4), buffered thermally treated sample stored at +4 °C; $B_{tt}$ (20), buffered thermally treated sample stored at −20 °C; $B_f$ (4), buffered filter-sterilized sample stored at +4 °C; $B_f$ (20), buffered filter-sterilized sample stored at −20 °C. Lower-case (a,b,c,d) letters are referred to values of the same incubation time, while capital (A,B,C,D) letters indicate statistically significant differences among values of the same series at different incubation time.

As regards the thermally treated samples, a significant increase in auxin/IAA concentration was observed after 30 days storage while it decreased in all samples at 60 days. The decrease was more evident in unbuffered samples. At 90-days storage, the highest auxin/IAA concentration was observed in unbuffered samples, with a +5% higher level. The temperature did not affect auxin/IAA concentration in both buffered and unbuffered samples (Table 1).

A similar trend was observed for samples sterilized by filtration (Table 1); auxin/IAA concentration increased from 0 to 30 days, with a higher increase in the unbuffered sample stored at +4 °C. After 30- to 60-day storage, a decrease in indole auxins concentration was observed in all filtered samples, except for sample $B_f$ (4), where the metabolite level remained almost constant. Sample $B_f$ (4) showed no significant differences in indole auxins level at 30-, 60- and 90-day storage. This implies that the treatments and storage conditions applied to this sample make auxin/IAA constant. At 90 days, the lowest indole auxins levels were found in $B_f$ (−20), while no significant differences related to storage temperature were observed in unbuffered samples. Data reported in Table 1 show that auxin/IAA levels were comparable, whichever pH and storage temperature was applied.

We used untargeted metabolomics to evaluate in more detail the effect of the pH and the sterilization treatment on the concentration of IAA, indole-3-acetamide (IAM), and other metabolites secreted by *Enterobacter* sp. strain P-36 in the VY medium. Chemical similarity enrichment analysis (ChemRICH) of statistically different annotated metabolites in samples stored at −20 °C for 30 days revealed some differences in the concentration level of IAA and IAM (Table 2).

**Table 2.** Effect of the thermal treatment (filtered vs. thermally treated unbuffered samples, $U_f$ vs. $U_{tt}$) or pH modification (unbuffered vs. buffered filtered samples, $U_f$ vs. $B_f$) on the indoleacetic acid-related compounds produced by *Enterobacter* sp. strain P-36. IAA = 3-Indoleacetic acid. IAM = Indole-3-acetamide. The data are referred to samples conserved for 30 days at $-20\ ^\circ$C.

| Treatment | Compound | PubChem ID | Smiles | *p*-Value | Effect_Size | Edirection |
|---|---|---|---|---|---|---|
| $U_f$ vs. $U_{tt}$ | IAA | 802 | C1=CC=C2C(=C1)C(=CN2)CC(=O)O | 0.0035 | 0.93 | down |
| | IAM | 397 | C1=CC=C2C(=C1)C(=CN2)CC(=O)N | 0.013 | 0.95 | down |
| $U_f$ vs. $B_f$ | IAA | 802 | C1=CC=C2C(=C1)C(=CN2)CC(=O)O | 0.0029 | 1.04 | up |
| | IAM | 397 | C1=CC=C2C(=C1)C(=CN2)CC(=O)N | 0.00014 | 1.10 | up |

These relative differences, comprised between 5 and 10%, agreed with the differences observed in measuring the total content of indole auxins using the spectrophotometric approach (Table 1). The same analysis indicated that a limited number of metabolites (belonging to six different clusters were significantly different ($p > 0.05$) between filtered and thermally treated samples (29 metabolites) and unbuffered and buffered samples (23 metabolites) (Table 3).

**Table 3.** Chemical Similarity Enrichment Analysis (ChemRICH) of *Enterobacter* sp. strain P-36 secreted metabolites whose concentration was altered by thermal treatment (filtered vs. thermally treated unbuffered samples, $U_f$ vs. $U_{tt}$) or pH change (unbuffered vs. buffered filtered samples, $U_f$ vs. $B_f$). The data are referred to samples conserved for 30 days at $-20\ ^\circ$C.

| Cluster Name | Cluster Size | *p*-Values | FDR | Altered Metabolites | Increased | Decreased | Increased Ratio | Altered Ratio |
|---|---|---|---|---|---|---|---|---|
| **$U_f$ vs. $U_{tt}$** | | | | | | | | |
| Amino Acids, Aromatic | 5 | $1.1 \times 10^{-15}$ | $5.6 \times 10^{-15}$ | 5 | 2 | 3 | 0.4 | 1 |
| Benzamides | 3 | $1.9 \times 10^{-3}$ | $2.1 \times 10^{-3}$ | 2 | 2 | 0 | 1 | 0.7 |
| Dipeptides | 6 | $0.1 \times 10^{-3}$ | $1.5 \times 10^{-4}$ | 5 | 3 | 2 | 0.6 | 0.8 |
| Indoles | 7 | $7.6 \times 10^{-10}$ | $1.9 \times 10^{-09}$ | 7 | 0 | 7 | 0 | 1 |
| Phosphatidic Acids | 3 | $2.0 \times 10^{-2}$ | $2.0 \times 10^{-2}$ | 2 | 0 | 2 | 0 | 0.7 |
| Phosphatidylcholines | 4 | $7.9 \times 10^{-9}$ | $1.6 \times 10^{-8}$ | 4 | 1 | 3 | 0.2 | 1 |
| Purines | 4 | $1.6 \times 10^{-9}$ | $2.6 \times 10^{-8}$ | 4 | 1 | 3 | 0.2 | 1 |
| **$U_f$ vs. $B_f$** | | | | | | | | |
| Amino Acids, Aromatic | 5 | $1.0 \times 10^{-7}$ | $4.7 \times 10^{-7}$ | 5 | 3 | 2 | 0.6 | 1 |
| Benzamides | 3 | $2.6 \times 10^{-8}$ | $2.4 \times 10^{-7}$ | 3 | 3 | 0 | 1 | 1 |
| Dipeptides | 4 | $4.0 \times 10^{-3}$ | $5.9 \times 10^{-3}$ | 3 | 2 | 1 | 0.7 | 0.8 |
| Indoles | 6 | $0.4 \times 10^{-4}$ | $0.9 \times 10^{-4}$ | 5 | 2 | 3 | 0.4 | 0.8 |
| Phosphatidic Acids | 3 | $2.1 \times 10^{-3}$ | $3.9 \times 10^{-3}$ | 2 | 1 | 1 | 0,5 | 0.7 |
| Phosphatidylcholines | 4 | $4.6 \times 10^{-3}$ | $5.9 \times 10^{-3}$ | 3 | 2 | 1 | 0.7 | 0.8 |
| Purines | 3 | $1.7 \times 10^{-2}$ | $1.9 \times 10^{-2}$ | 2 | 2 | 0 | 1 | 0.7 |

These changes could be explained considering the thermal or pH stability of some of these compounds (indoles, purines, and aromatic amino acids) and the possible effect of the temperature and the pH on the stability and activity of secreted enzymes that could be involved in the hydrolysis of conjugated molecules or degradation/modification of specific metabolites.

## 4. Discussion

Several *Enterobacter* strains have proved so far to exhibit an ability to produce high auxin/IAA levels, which is one of the main direct mechanisms to promote plant growth [20–27]. Bacterial strains, growth medium formulation (e.g., carbon and nitrogen source, concentration of indole auxins precursor), and growing conditions (e.g., agitation speed, temperature, and pH) generally affected indole auxins production, and evidence showed that this is applicable for other strains affiliated with Enterobacteriales (*Enterobacter cloacae/ludwigii*), as well.

In this study, three different growth media were investigated in preliminary experiments in shake flasks. It emerged that *Enterobacter* sp. strain P-36 was a better-performing auxin/IAA producer when cultivated in VY + Trp (4 mM) than in YES or LB media. In

detail, the lowest auxin/IAA concentration was obtained in YES medium (31.4–123.9 mg IAA$_{equ}$ L$^{-1}$), whichever fermentation duration was applied (24, 48, and 72 h), while the highest levels occurred when cells were grown on VY medium amended with tryptophan (219.4–459.3 mg IAA$_{equ}$ L$^{-1}$).

Data presented in Figure 1 showed that in all tested conditions, while differences were observed in the auxin and biomass yield profiles, the auxin-specific productivity did not significantly change between 48 and 72 h. These data indicated that the auxin production was uncoupled by the biomass production and suggested that, in *E. ludwigii* strain P36, the production of this metabolite represented an energy-spilling reaction typical of a rate-yield tradeoff phenotype [28].

Comparing the auxin/IAA level obtained in this work with the data reported in other studies [21,22] for a range of *Enterobacter* strains cultivated in shake flasks, great variability was observed. The levels of IAA (219.4–207.9 mg L$^{-1}$) produced by *Enterobacter* sp. strain P-36 upon fermentation in VY + Trp medium at 24 and 48 h were comparable to the concentration (282.4 mg L$^{-1}$) reported for *Enterobacter* sp. strain DMKU-RP206 after 24-h cultivation in TSB medium [22]. Additionally, IAA/auxin production (459.3 mg L$^{-1}$) by strain P-36 after 72-h growth in VY + Trp medium was comparable to the maximum amount of auxin/IAA (448.5 mg L$^{-1}$) observed for *Enterobacter* sp. DMKU-RP206 at 10 days of incubation [22]. The maximum auxin/IAA level produced by *Enterobacter* sp. strain p-36 in this study was, on the other hand, 7-fold lower than the levels (3158.8 mg L$^{-1}$) obtained by cultivating strain DMKU-RP206 in shake flasks with an animal-based medium formulated with an optimal concentration of lactose (0.85%), yeast extract (1.3%), L-tryptophan (1.1%), and NaCl (0.4%), as well as at optimal pH (5.8), incubation temperature (30 °C) and agitation speed (200 rpm) [22]. However, our results demonstrated that it was possible to obtain high auxin/IAA levels using a medium free of animal-derived ingredients that could be very attractive in applying this product in eco-friendly agriculture.

The scale-up of *Enterobacter* sp. strain P-36 cultivation in a 2-L stirred tank fermenter allowed obtaining, at 48-h fermentation, an auxin/IAA production (412.49 mg IAA$_{equ}$ L$^{-1}$) comparable to that observed in shake flasks at 72 h. However, the auxin/IAA concentration determined upon strain P-36 cultivation in the bench-top fermenter was 12-fold lower than observed for *Enterobacter* sp. DMKU-RP206 was cultivated in a bench-top fermenter for seven days. The auxin/IAA production observed in this study was also 7-fold lower than reported by Nutaratat and Srisuk [21] in a study where strain DMKU-RP206 was grown in 2- and 15-L stirred tank fermenter with sweet whey yeast extract (SY) L-tryptophan medium [21].

Auxin/IAA production by *Enterobacter* sp. strain P-36 upon cultivation in both shake flasks and bench-top fermenter was, on the other hand, by far higher than the levels reported in the literature for *Enterobacter cloacae* strain H3 and *E. cloacae* MG00145 [25,26]. *E. cloacae* strain H3 produced, in fact, 12.28 mg L$^{-1}$ of auxin/IAA in nutrient broth medium at pH 6 and containing 0.5 mM L-Tryptophan [25]. An auxin/IAA yield ($\approx$18 mg L$^{-1}$) comparable to strain H3 was observed in *E. cloacae* MG00145 after 8-day growth at 37 °C and pH 7 in a culture medium formulated with 0.5% sucrose and 0.1% calcium nitrate as optimal carbon and nitrogen source, respectively [25].

*Enterobacter* sp. strain P-36 produced auxin/IAA levels higher than other PGPR strains. A total of 120.5 ± 0.9 and 157.67 mg IAA$_{equ}$ L$^{-1}$ were reported for *Pantoea agglomerans* strain C1 cultivated in YES medium, in the presence of both sucrose and yeast extract at a final concentration of 0.5% (*w/v*) [13], and in LB medium amended with tryptophan (0.4 mM) at 30 °C temperature and 180 rpm agitation speed [14], respectively.

It might be therefore interesting to investigate auxin/IAA production by *Enterobacter* sp. strain P-36 in other media and optimizing its production using statistical methods (e.g., response surface methodology).

This study offered interesting insights into the possibility to develop more sustainable fermentation processes for auxin/IAA production by PGPR strains, in terms of ingredient origin (animal vs. plant-based medium components), costs, and fermentation time. The

identification of the VY medium as the one allowing a higher production of auxin/IAA by *Enterobacter* sp. strain P-36 than other media opened the way to the use of GMO-free vegetable peptones in the formulation of PGPR growth media. The successful application of the plant-based component was of great interest, as it met the recent needs of the bio-industry market to identify alternative peptones that (i) were free from animal-derived components, (ii) had similar nutritional benefits compared to soy and milk proteins, and (iii) could provide a highly nutritious general-purpose range of media for the growth of bacteria and fungi.

The study also contributed to the identification of approaches to save cost and time in the fermentative process. In bench-top fermentation of *Enterobacter* sp. strain P-36, an increase in auxin/IAA production was observed at the increase in bacterial inoculum size, in keeping with the findings of Nutaratat and Srisuk [21]. It emerged that when increasing the inoculum from 0.5 to $1.0 \times 10^9$ cell/mL, *Enterobacter* sp. strain P-36 produced $427.9 \pm 8.1$ mg $IAA_{equ}$ $L^{-1}$ in 24 h instead of $275.9 \pm 5.0$ mg $IAA_{equ}$ $L^{-1}$. Optimization of the cultivation conditions determined a reduction in the fermentation time from 72 (shake flask cultivation) to 24 h (bioreactor cultivation) and an increase in the auxin volumetric productivity from 6.4 to 17.2 mg $[IAA_{equ}]$/L/h. As a result of the reduction in fermentation time, raw material and production costs could be also reduced. These results, therefore, have important benefits at both economic and sustainability levels.

As regards the effect of storage parameters on the stability of crude preparations of auxin/IAA, a significant increase thereof emerged after 30 days with respect to day zero. This was explained by the fact that auxin can be bound to amino acids and sugars [29,30], therefore, these conjugated forms could be a reservoir in plants [31]. It is also known that in plants, auxins conjugated to L-aspartic (indol-3-acetyl-L-aspartic acid, IAAsp) and L-glutamic acid (indol-3-acetyl-L-glutamic, IAGlu) are not hydrolyzed, consequently, their formation determines irreversible sequestration of the active auxin [32,33]. In addition, auxins could be found conjugated in a reversible way to glucose by the action of UDP-glucosyltransferase which forms the indole-3-acetyl-glucosyl ester (IAGlc) [34,35]. Based on that, we could thus assume that strain P-36 produced conjugated auxin forms, that were hydrolyzed either completely or partially during storage, and an increase in auxin/IAA concentration hence occurred. This increase was higher in the unbuffered sample than in the buffered sample at pH 9.

The analysis of the effect of storage parameters on indole auxins concentration points out that indole auxins levels were comparable, whichever pH and storage temperature was applied. Therefore, medium pH did not require correction; samples could be stored at 4 °C; and microbiological stabilization could be accomplished by whichever technology, e.g., filtration or thermal treatment. In addition, no significant differences could be observed between auxin/IAA concentrations at 60 and 90 days. This means that these metabolites were stable for at least 90 days.

Results from untargeted metabolomics supported the conclusion that the storage parameters had a minor effect on the concentration of IAA and IAA-related compounds as well as on the profile/concentration of other major compounds secreted by *Enterobacter* sp. strain P-36.

## 5. Conclusions

*Enterobacter* sp. strain P-36 proved to be a major auxin producer via fermentation. In the context of green chemistry, the use of a growth medium formulated with a vegetable peptone, instead of the animal-based tryptone, is of interest. It showed economic and environmental advantages that make the fermentative process exploitable on a large scale for the formulation of plant-biostimulants applicable to organic and vegan agriculture.

The results obtained from the experiments in the 2-L stirred tank fermenter showed that the developed protocol was applicable for a process scale-up. In detail, the addition of the IAA-precursor at the inoculum and the identification of the cell volume to inoculate

the medium allowed high auxin/IAA levels and a parallel reduction of the fermentative process duration with respect to the shake flasks.

The protocol developed for product storage showed that *Enterobacter* sp. strain P-36 produces, in VY medium, auxin in free and bound form. The latter hydrolyzed by the first 30 days of storage, whichever medium pH and temperature were applied. Indirectly, it could be also concluded that the amount of auxin estimated spectrophotometrically was underestimated with respect to the overall amount of auxin produced by the strain.

Storage experiments showed that the broths containing the auxins could be stored at +4 °C for at least 90 days, with no negative effects on auxin/IAA levels. Microbial stabilization could be performed by both filtration and thermal treatment at 121 °C, without significantly modifying the properties of the broth.

To conclude, the biological system developed via fermentation in this study is suitable for auxin/IAA production and application to conventional, as well as organic and bio-vegan agriculture.

**Author Contributions:** Conceptualization, F.L. and M.R.; methodology, F.L. and M.R.; software, F.L., P.B. and M.R.; validation, F.L., P.B. and M.R.; formal analysis, M.R.; investigation, F.L., F.M., P.B., V.M. and M.R.; resources, F.L., F.M. and M.R.; data curation, F.L.; writing—original draft preparation, F.M.; writing—review and editing, M.F, F.L. and M.R.; visualization, F.L.; supervision, M.R.; project administration, M.R. and V.C.; funding acquisition, M.R., V.C. and P.B. All authors have read and agreed to the published version of the manuscript.

**Funding:** This research received no external funding.

**Data Availability Statement:** All datasets generated for this study are included in the manuscript.

**Acknowledgments:** This work was supported by internal resources of the participating institutes.

**Conflicts of Interest:** The authors declare no conflict of interest. The funders had no role in the design of the study; in the collection, analyses, or interpretation of data; in the writing of the manuscript, or in the decision to publish the results.

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
