# Peer review of "Production of Indole Auxins by Enterobacter sp. Strain P-36 under Submerged Conditions"

_fermentation, doi:10.3390/fermentation7030138_

Round 1

Reviewer 1 Report

Dear Authors,

New agricultural techniques development is essential to obtain sustainable agriculture to produce food for humankind. The use of bacteria, and its by-products, will be a key tool to reach this objective. The manuscript presents an interesting work about fermentation procedures to obtain auxin phytohormones throughout the optimization of Enterobacter sp. P36 culture.

The design of the manuscript is correct, with a good text redaction. I only want to ask about the explanation of the observed auxin levels in the storage system, why are there changes in auxin concentration? It is not strange a decrease in concentration, but yes the increase appeared in some treatment. Also, the reader will expect to find a clear protocol, I think that this point should be clarified, defining clearing the steps and giving an explanation for the observed results.

Best regards

Author Response

We would like to thank the Reviewer for the time spent reviewing our manuscript and the insightful comments helping us improving the article.

Below, the authors have tried to answer the questions and reply to the referee comments, point by point.

As requested by Reviewer 1, the Introduction section was expanded (lines 68-77 of the revised manuscript), and we schematized the procedure used for the preservation of the auxin to facilitate the understanding of the protocol (lines 174-176 of the revised manuscript).

A possible explanation for the increase in the auxin level at 30 days of storage was already present (lines 556-567 of the revised manuscript). We believe that the metabolomics data analysis that we included in the revised manuscript will help to clarify our observations.

Reviewer 2 Report

Dear Editor,

The paper Development of a fermentative process for production of indole 2 auxins by Enterobacter sp. strain P-36 3, presented by Luziatelli et al., aimed to describe a bioprocess for the production of indole auxins. The authors have highlighted the importance of producing plant regulators such as IAA and provided some examples on the cultivation of Enterobacter sp. P26. The authors also tested the viability of the auxin produced.

General comment

While the topic is relevant given the need for biobased products for agriculture, the experimental approach and the results presented are flawed. This reviewer considers the work has merit but the manuscript must be improved to present a clearer description of the fermentative process the authors claimed had developed. Overall, for a manuscript submitted to Fermentation, there is limited description of the actual fermentation process and the result show no data on growth curves, growth rate, yields and productivities. In addition, the reported values of auxin production should be corroborated using a more powerful technique like HPLC or LCMS, to at least determine which molecules are being produced. The authors should describe clearly how the process was developed or at least describe the rationale behind the conditions tested. Finally, I strongly suggest to re-organise the manuscript in a logical sequence: experiments in flask, experiments in reactors, evaluation of the auxins, and report growth curves and is available the consumption of tryptophan. 

Specific comments:

Line 88 – update reference style

118 – ‘blowing sterile air” sounds informal. Could be changed for “maintaining a constant air flow rate”.

Line 167. The authors indicate results appear on table 1, but this is confusing since Table 1 shows effects of pH temperature. Figure 1 shows auxin titres and the results are confusing. There is no indication of what Induction time means. Also, where are the growth profiles? Authors should provide a clearer picture of the growth of Enterobacter and provide the corresponding comparison and correlation between growth rates, yields and productivities. Also, why the saline medium was not tested in shake flasks? Why the saline medium was never mentioned? Also the authors should test a control without tryptophan 

Line 185. The figure is confusing and more info is required  in the figure caption.  Description must include the number of replicates and indicate what a, b, c, A, B, and C are.

Line 193. The authors make reference to optimum conditions, however those are never described. Also, if these are optimum condition, what other conditions were tested?

Line 201. This reviewer finds several flaws in the experimental approach for measuring growth. The authors claim that they used pH as an indirect measure of cell growth because the low solubility of the peptones interferes with the OD measurements. Still, the describe that the initial OD was  0.2. While it is understandable that complex media can interfere with OD measurements, still there are other methods the authors could’ve used to evaluate cell growth (CFU per mL in limiting dilutions plated on agar plates, for instance). Dissolved oxygen can also provide a better idea on cell growth. Finally, using pH as proxy for cell growth on complex media rich in peptides will only indicate increase on pH due to ammonia release from peptide hydrolysis. It is also not clear why cells were grown in a  reactor and then transferred to flasks for production of IAA

Line 249. At this stage, the whole results are still unclear and there is not clear rationale behind each experiment. Also, the data presented does not follow a clear order, which makes the manuscript hard to read.

Line 301. Which units are correct? mg/mL or ug/mL.

Line 307. Why did the authors not tested these conditions with their strain? Why pH was not controlled in your fermentations?

Line 319. Authors keep mixing units. While the authors compared the titres with other works, yields and productivities are better parameters to use.

Line 360. Authors should perform more detailed analysis in they samples to identify which molecule is being produce and whether or not is being degraded (HPLC, LCMS)

Author Response

We would like to thank the Reviewer for the time spent reviewing our manuscript and the insightful comments helping us improving the article.

Below, the authors have tried to answer the questions and reply to the referee comments, point by point.

Unfortunately, after the formatting changes introduced by the MDPI editorial office, the position of Table 1 has been changed (from page 9 to page 4), and the reading of the text was seriously compromised.

Figure 1 was modified, and we included information on the effect of the culture medium on the biomass yield and auxin-specific productivity that we think can provide a more detailed picture of the fermentation process (page 6).

We also modified Figure 2, including direct (CFU/mL) and indirect (pH evolution) measures of the cell growth (page 7).

Finally, we included data from an untargeted metabolomics analysis that support the spectrophotometric data included in the manuscript (lines 423-471, Table S1 and S2).

Line 88 (94 in the revised manuscript): the reference style was changed.

Line 118 (137 in the revised manuscript). The sentence was changed according to the Reviewer’s suggestion.

Line 167: the position of Table 1 was changed. Figure 1 was changed. The word “induction” was substituted with “incubation.” The text of “subsection 3.1” was extensively revised (lines 207-291). Data on biomass yield and auxin-specific productivity have been included in the revised manuscript (Figure 1). To avoid confusion about the YES medium, we omitted the attribute “saline” (line 93). The reference to the experimental test without tryptophan has been introduced in the revised manuscript (lines 210-212).

Line 185 (284 in the revised manuscript): Figure 1 and its caption have been changed.

Line 193 (298): the sentence has been changed following the Reviewer’s suggestion.

Line 201 and 236 (306 and 359 in the revised manuscript): the sentences have been changed to address the Reviewer’s comments. The inoculum was expressed as the initial number of cells per volume unit. Figure 2 was revised as mentioned before.

Line 249 (372): the revisions introduced in the text should facilitate the reading of the manuscript.

Line 301 and 319: all the data have been converted in “mg L-1”. We agree with the Reviewer on the use of the yield and productivity as comparison metrics. Unfortunately, several authors working on the IAA production do not provide these data, and they do not give enough information to extrapolate them.

Line 307: the medium utilized by Nutaratat et al. contains animal-derived ingredients, and the main goal of this work is to use a medium free of animal-derived ingredients (lines 501-504). The pH was not controlled in our fermentations because we think that the optimization of the fermentation process (temperature, pH, DO level, rotation speed) should be done using a rational statistical approach. We aim to use the Response Surface Methodology to deepen this topic, and we are planning to present the results in a separate paper.

Line 360: the question has been addressed with the metabolomics data analysis presented the subsection 3.3 (lines 423-471, Table S1 and S2).

Round 2

Reviewer 2 Report

Dear Editor,

Dear authors,

I have read the revised version and the responses of the authors to my observations.

While the authors have addressed the comments I made on their work, I still have serious concerns about the presentation and the experimental approach. Therefore, I intend to provide useful feedback to the authors to generate a research output that meets the standards of a scientific journal such as Fermentation. Below you will find my comments on the responses given by the authors, which I still consider require substantial revision:

  • The new figure 1 introduces additional questions. While in the methodology, the authors describe that for flask experiments, they use 25 mL cultures and sample 10 ml for auxin production after 24 h, how is that the authors reporting data up to 72 h? How can they maintain 15 mL cultures for 72 h and still have samples for analysis? Also, in Figure 1 panel A it shows that only cells grown in YES medium. Data for LB and VY media indicate no growth or reduction of biomass (errors also show no differences across time on these media). If those results are correct, how is it that the authors report the higher production yields on VY media? If these results are correct, the authors must demonstrate the viability of the cells?
  • I am still not convinced of using pH as a proxy for growth. While the trends are similar, I would still recommend removing the pH data and onlu use CFU or OD values. Still, the authors do not provide basic parameters such as growth rates and yields. While the authors report yields, the data correspond to final titres.
  • While authors claimed that yields and productivities are not reported in other works, they should be aware that the title of their work is “Development of a fermentative process…”. Therefore, the corresponding parameter for a fermentative process must be provided.
  • The authors claimed that ‘the optimization of the fermentation process (temperature, pH, DO level, rotation speed) should be done using a rational statistical approach. We aim to use the Response Surface Methodology to deepen this topic, and we are planning to present the results in a separate paper. ’. Still, section 2.3 claims to be the Optimization of indole auxin production in 2-L stirred tank fermenter. This information is either incorrect or misleading. Also, in this section, the authors aimed to evaluate the induction time by taking samples from the reactor and growing them in flask afterwards. While the rationale is somewhat clear, the results cannot be presented as part of an optimisation process. The correct way to test this is by inducing the culture in the reactor.
  • The data for the MS analysis should be present in the main manuscript. Also, the MZ spectra comparing standards and samples should be included to make the results sounding.

Author Response

To respond to the recommendation and major criticism of the Reviewer, we have

  • changed the title of the manuscript and removed any reference to the “fermentation process”.
  • changed the title of paragraph 2.3.
  • removed all the experimental sections on the time of induction (data discussed in former Figure 1 and Figure 2 and the experimental procedure described in paragraph 2.3).
  • transferred the supplementary tables in the main document.
  • provided additional information on MS analysis.

In details:

Question 1 about figure 1. “How can they maintain 15 mL cultures for 72 h…”

We have specified in the text (Line 131-133) that, in our shake flask experiments, we try to minimize common errors due to variation in the medium-to-flask volume ratio. If the aliquot withdrawn is not negligible compared to the total volume of liquid, you will determine an uncontrolled variation in the aeration rate with a direct effect on the metabolic state of the bacteria.

For this purpose, we have inoculated 9 flasks for each treatment, and triplicates of these flasks were sacrificed at each time point.

Question 2 about Figure 1.  “cell growth… viability of the cells?”

The paragraph was changed and the differences in the cell growth were discussed (line 237-243). Our data indicate that, in VY medium, the auxin-specific productivity remained constant between 48 and 72 hours, and the biomass yield increases between 24 and 72 hours. The auxin production requires energy, if the cell viability would be impaired, we would not observe an increase in the auxin-specific productivity between 24 and 72 hours.

I am still not convinced of using pH as a proxy for growth.

To avoid any confusion, We have removed all the experimental sections on the time of induction

Authors should be aware that the title of their work is “Development of a fermentative process…”.

To avoid any misleading message, the title of the manuscript has been changed.

Section 2.3 “Optimization of indole auxin production in 2-L stirred tank fermenter”. This information is either incorrect or misleading.

The title has been changed (line 134).

The data for the MS analysis should be present in the main manuscript.

The data have been moved in the main text (Table 2, line 417; Table 3, line 439). The data requested on the methodology have been introduced in the Materials and Methods section (lines 204-208).

Round 3

Reviewer 2 Report

Dear Editor,

Dear authors,

I have read the revised version and the manuscript and the and the responses provided. I want to thank both the editor and the authors for the opportunity to contribute in the revision of this manuscript. The authors have addressed the comments and concerns I made, and the current version of the manuscript provides a clearer description of the authors’ research. 

While I acknowledge the improvement in the manuscript, there is still a major concern the authors should address in the discussion section regarding the results reported in Figure 1. In the manuscript (line 211), the authors claim that “A significant increase in the biomass yield was observed in YES and VY medium between 24 and 72 h.” However, Figure A actually shows that growth is only seen in YES media. LB media cultures decreased the biomass concentration, and VY data indicates no growth since the error bars overlap. The authors should elaborate on why auxins production is present while cell growth is arrested. I agree with the response of the authors “The auxin production requires energy, if the cell viability would be impaired, we would not observe an increase in the auxin-specific productivity between 24 and 72 hours.”. Still, authors should elaborate on why and how the bacteria can generate auxins while not growing. Also, if the cells are not growing, how are they generating energy, and how are they balancing the redox?  This should be clearly described and discussed, as the results from that experiment led the authors to the selection of VY media for the next experiments.

Regarding the MS results, I would encourage the authors to include some spectra plots for the samples compared with the standards to make the claims more sound.

Finally, there are still a few typos to be corrected.

Author Response

Responses to the recommendation and major criticism of the Reviewer

Reviewer 2’s concerns about Figure 1.

After carefully considering the Reviewer’s comments and rereading our manuscript, we have

  • Modified the sentence “A significant increase in the biomass yield was observed in YES and VY medium between 24 and 72 h” as follows: “Between 24 and 72 hours, the biomass yield increased by 2.8-fold (from 1.41±0.1*1013to 3.93±0.2*1013 CFU L-1) in YES medium and 1.8-fold (from 3.2±0.8*1012 to 5.9±0.5*1012 CFU L-1) in VY medium, respectively” (lines 198-200).
  • Added the following statement in the Discussion section: “Data presented in Figure 1 show that in all tested conditions, while differences are observed in the auxin and biomass yield profiles, the auxin-specific productivity does not significantly change between 48 and 72 hours. These data indicate that the auxin production is uncoupled by the biomass production and suggest that, in E. ludwigii strain P36, the production of this metabolite represents an energy-spilling reaction typical of a rate-yield tradeoff phenotype [28]” (lines 361-366).

Reviewer’s comment on spectra plots

Regarding the Reviewer’s suggestion on the MS results, we think that the inclusion of the spectra plots in the Supplementary material would not provide helpful information “to make our claims more sound.” We leave it, however, for the Editor to decide how to proceed from here.

Typos to be corrected

We have corrected typos present in the manuscript.